# Wave-wise comparative genomic study for revealing the complete scenario and dynamic nature of COVID-19 pandemic in Bangladesh

Ishtiaque Ahammad[1⊙], Mohammad Uzzal Hossain[1⊙], Anisur Rahman[1⊙], Zeshan Mahmud Chowdhury[1], Arittra Bhattacharjee[1], Keshob Chandra Das[2], Chaman Ara Keya[3], Md. Salimullah[2]*

1 Bioinformatics Division, National Institute of Biotechnology, Dhaka, Bangladesh, 2 Molecular Biotechnology Division, National Institute of Biotechnology, Dhaka, Bangladesh, 3 Department of Biochemistry and Microbiology, North South University, Bashundhara, Dhaka, Bangladesh

⊙ These authors contributed equally to this work.
* salim2969@gmail.com

**Data Availability Statement:** All relevant data are within the paper and its Supporting Information files.

## Abstract

As the COVID-19 pandemic continues to ravage across the globe and take millions of lives and like many parts of the world, the second wave of the pandemic hit Bangladesh, this study aimed at understanding its causative agent, SARS-CoV-2 at the genomic and proteomic level and provide precious insights about the pathogenesis, evolution, strengths and weaknesses of the virus. As of Mid-June 2021, over 1500 SARS-CoV-2 genomesequences have been deposited in the GISAID database from Bangladesh which were extracted and categorized into two waves. By analyzing these genome sequences, it was discovered that the wave-2 samples had a significantly greater average rate of mutation/sample (30.79%) than the wave-1 samples (12.32%). Wave-2 samples also had a higher frequency of deletion, and transversion events. During the first wave, the GR clade was the most predominant but it was replaced by the GH clade in the latter wave. The B.1.1.25 variant showed the highest frequency in wave-1 while in case of wave-2, the B.1.351.3 variant, was the most common one. A notable presence of the delta variant, which is currently at the center of concern, was also observed. Comparison of the Spike protein found in the reference and the 3 most common lineages found in Bangladesh namely, B.1.1.7, B.1.351, B.1.617 in terms of their ability to form stable complexes with ACE2 receptor revealed that B.1.617 had the potential to be more transmissible than others. Importantly, no indigenous variants have been detected so far which implies that the successful prevention of import of foreign variants can diminish the outbreak in the country.

## Introduction

Severe Acute Respiratory Syndrome Coronavirus-2 (SARS-CoV-2), the causative agent of Coronavirus Disease-2019 (COVID-19) has already infected more than 177,000,000 people and caused over 3,800,000 deaths till mid-June, 2021) [1]. Since the influenza outbreak of 1918

**Funding:** The authors received no specific funding for this work.

**Competing interests:** The authors have declared that no competing interests exist.

COVID-19 is the biggest pandemic of zoonotic origin that we are facing at a global scale [2]. The first wave of the pandemic has passed and subsequent waves have already started in many countries [3–7]. Insights regarding the transmission and evolution of the virus during these waves are essential to break the chain of infections [8, 9]. Genomic data can provide some of these crucial insights which can help make pragmatic public health policies [10, 11]. Besides, genomic surveillance can deliver a deep understanding of the virus' mechanism of survival and reduce fatality during new waves of infection [11–14].

The onset of SARS-CoV-2 occurred in Wuhan, Hubei Province, China in December, 2019 [15–17]. Initially, clinicians diagnosed this disease as virus-induced pneumonia based on blood tests and chest radiographs. Later, genomic and phylogenetic data analysis led to the recognition of the pathogen as a member of the *Coronaviridae* family [18]. *Coronaviridae* family encompasses the largest known enveloped, single stranded RNA viruses with a genome size ranging from 25–32 kilo base pairs (Kb) [19, 20]. The family is divided into two subfamilies, the *Coronavirinae* and the *Toronavirinae*. The subfamily *Coronavirinae* is further organized genotypically and serologically into 4 genera: α, β, γ, and δ-CoVs [21]. The *betacoronavirus* genus is comprised of the Severe Acute Respiratory Syndrome (SARS)-CoV which had been identified for the first time in 2002–2003 and the Middle East Respiratory Syndrome (MERS)-CoV in 2012. The genome sequences of SARS-Cov-2 has a 79.6% identity with SARS-CoV/ SARS-CoV-1 and 67.06% identity with MERS-CoV, indicating that they belong to the *betacoronavirus* genus [22]. All human coronaviruses are considered to be of zoonotic origin, with Chinese bats being the most likely host for SARS-CoV-2 [23–25]. Genetically, about 96% identity was observed between SARS-CoV-2 and bat coronavirus (BatCoV RaTG13) [18]. However, since bat habitats remain distanced from human life, an intermediate animal such as pangolin might have acted as an intermediate shuttle before transmitting to its human hosts [26–30].

The Chinese Center for Disease Control and Prevention (CDC) primarily suggested the Huanan local seafood market as the origin of the COVID-19 outbreak [31]. Despite this claim, none of the animals in the area were tested positive for the virus. This indicated that the virus had already moved out of Wuhan, long before the outbreak came under spotlight. Since then the control of viral transmission through non-therapeutic interventions suggested by the World Health Organization (WHO) had been attempted [32]. However, the violation of these preventive measures and absence of proper antiviral therapeutics and vaccinations led to an uncontrollable global transmission of the disease. The virus proliferated rapidly both inside and outside of China and finally reached each and every county of the world. In March 2020, the disease was declared as a global pandemic by the World Health Organization (WHO) [15]. Although, at the beginning of the pandemic, the intensity of the disease was higher in the Europe and the America but later it also spread to Asian and South-East Asian countries [33–35].

Previously, the world went through three waves of the deadly Spanish flu until it subsided in 1919 while the second wave being the deadliest. The reason behind this fatal phenomenon was the rapid dispersion of the virus to every corner of the world [36]. A similar pattern can be observed in the case of COVID-19. By late 2020s and early 2021, a resurgence of infections was experienced by most countries including the United States, Brazil, Belgium, France, UK, Germany, as well as most of the Asian countries [37–39]. Remarkably India, which survived the first wave relatively unscathed, is currently suffering from a spine-chilling situation with a higher mortality rate than most other countries seeing more than 2000 deaths per day [40].

A well-established fact is that all viruses undergo genetic drift over time due to selection pressure and give rise to a number of variants that challenge any pandemic response [41, 42]. Therefore, understanding the current variants are crucial in restricting the mode of

transmission and developing new therapeutics against them. Multiple variants have been identified around the world so far including B.1.1.7 [43], B.1.351 [44], P.1 [45], B.1.427/B.1.429 [46] and B.1.617 [47]. The B.1.1.7 variant was first detected in the United Kingdom around September, 2020. Three different types of mutations were observed in this variant which were present in the receptor binding domain of the spike protein, the 69/70 deletion and the P681H mutation near the S1/S2 furin cleavage sites. The alpha or kent variant turns out to be mutating again. In December 2020, B.1.351 was spotted as the predominant variant in South Africa [48]. The variant, sharing some mutations with B.1.1.7 also had multiple mutations in their spike proteins such as K417N and E484K [49]. The P.1 variant was first identified in Japan in a few travelers coming from Brazil in early January, 2021 [50]. B.1.427 and B.1.429 variants were first detected in California in February 2021 [46]. B.1.617.2 is the daunting variant of coronavirus that originated in India and has been circulating globally in at least 62 countries including the United States and United Kingdom [51]. About 70% of the genome sequences submitted from India to GISAID constitute this variant. The major mutations in the delta variant includes substitution in the amino acid sequences of the spike protein [52, 53].

Bangladesh, being one of the most densely populated countries of the world with over 160 million people and sharing a porous border with India, remains one of the most vulnerable countries for the second wave of the COVID-19 pandemic. The country with limited resources and scarce healthcare facilities experiences major challenges while combating this transmission. The first case of this virus in the country was confirmed in two men coming from Italy and a female relative by the Institute of Epidemiology, Disease Control and Research (IEDCR) on March 7th, 2020 [54] Although many Bangladeshi citizens came from Wuhan beforehand, they were reported to be negative for SARS-CoV-2. As a response, the Bangladesh government took a number of preventive measures including nationwide lockdowns, imposing restrictions on international flights, strengthening of screening procedures, and shutting down of educational institutions and so on [55]. Despite several rounds of lockdowns, the rate of infections continued to reach high levels. Correspondingly, it became the second most affected country in Southeast Asia. Near the end of the first wave, it began to drop gradually since November 2020. Although the rate declined to its lowest during January and February, 2021, the cases began to rise again [56].

The first complete genome sequencing of the SARS-CoV-2 in Bangladesh was announced by the Child Health Research Foundation on 12th May, 2020 [57]. Soon after, the National Institute of Biotechnology announced the sequencing of SARS-CoV-2 genome by Sanger sequencing method [58]. The SARS-CoV-2 genome sequencing effort in Bangladesh flourished afterwards and as a result, 1569 genomes have been sequenced by June 6, 2020.

The goal of this study was to probe all these sequences and find some crucial answers regarding the genomic evolution of the virus, predominant variants, difference between the first and the second wave and so on which would make it easier to comprehend the trajectory of the pandemic and suggest appropriate counter measures.

## Materials and methods

### Retrieval of the SARS-CoV-2 genome sequences

Genomes of SARS-CoV-2 isolates were retrieved from the Global Initiative on Sharing All Influenza Data (GISAID) database (www.gisaid.org) [59]. Isolates collected since the beginning of the COVID-19 pandemic till 31 Jan 2021 were considered as wave-1. (S1 File) and those collected between Feb 1, 2021 and Jun 6, 2021 were considered as wave-2 samples (S2 File).

## Wave-1 and wave-2 mutation analysis

The Wuhan genome reference sequence (NC_045512.2) was retrieved from NCBI GenBank [60]. A GFF3 annotation file of the reference sequence (**S3 File**), generated by Giorgi was used for extracting the genomic coordinates of SARS-CoV-2 proteins [61]. The sequences from wave-1 and wave-2 were aligned separately against the reference sequence using the NUCMER (version 4.0.0rc1) command line tool [62]. A SARS-CoV-2 annotation algorithm, developed by Mecatelli and Giorgi [61], was employed to convert the outputs of alignments into lists of mutational events. Frequency and the rate of mutation per sample was calculated. All the SARS-CoV-2 sequences from both waves were classified based on the type of mutation. Specific coordinates of the mutations on the SARS-CoV-2 genome were also identified. Finally, alterations in the proteome of SARS-CoV-2 as a result of genomic variation were investigated. **S4 File** contains a detailed report on this mutation analysis for both COVID-19 pandemic waves in Bangladesh.

## Clade and variant analysis for wave-1 and wave-2

For this analysis, both complete and incomplete sequences for wave-1 and wave-2 in the Bangladesh region were obtained from GISAID. The number of sequences for different clades was counted directly from this database. Assignment of different lineages for each sample from both waves was performed by pangolin (version v3.0.5, lineages version 2021-06-05) web server (https://pangolin.cog-uk.io/). For all sequences, Greek Alphabet names of relevant lineages, as well as their classes (VOC for variants of concern, VOI for variants of interest and Unclassified for other variants), were also ascribed. Percentage of occurrences for different clades and top ten variants were calculated via R commands. Finally, comparative plots were generated by using the ggplot2 package in R [63] to describe the distribution of SARS-CoV-2 clades and variants in Bangladesh.

## Analysis on selection pressure and molecular phylogeny of the variants

Selection pressure on the SARS-CoV-2 proteome was explored to find out both the positively and negatively selected variants in between wave-1 and wave-2. More specifically, the role of selection pressure on spike protein variants in light of the samples of Bangladesh was investigated. Frequency (%) of top ten spike protein variants was compared in the context of wave-1 and wave-2. Molecular phylogeny was examined using the Nexstrain tool integrated in GISAID database [64].

## Molecular docking between spike protein and ACE2

The reference sequence of the Receptor Binding Domain (RBD) of SARS-CoV-2 spike glycoprotein (S) was taken from UniProt (https://www.uniprot.org/) (UniProt ID: P0DTC2) and was manually mutated to generate the sequence of the variants B.1.1.7, B.1.351, and B.1.617 which were most common in Bangladesh. 3D models of all the sequences were built using Robetta [65]. The structure of the human ACE2 receptor was extracted from the RCSB PDB (PDB ID: 6M0J). Docking between S-RBD and ACE2 was conducted by GalaxyTongDock_A server [66]. Following protein-protein docking, the generated models with the highest docking scores and cluster size were selected and submitted to PROtein binDIng enerGY prediction (PRODIGY) to calculate the binding affinity of the protein-protein complexes at physiological temperature (37˚C) [67].

## Molecular dynamic simulation of ACE2-spike protein complex

In order to evaluate the evaluate the stability of the complex between the ACE2 receptor and the SARS-CoV-2 Spike protein (Reference and the variants B.1.1.7, B.1.351, and B.1.617)

under physiological conditions, 50 ns molecular dynamics simulation was executed with GROningen MAchine for Chemical Simulations aka GROMACS (version 5.1.1) [68]. The GROMOS96 43a1 force-field was used for the simulation [69]. 300 K temperature, pH 7.4, and 0.9% NaCl solution was used to define the physiological condition of the system. A dodecahedral box with its edges at 1 nm distance from the protein surface was drawn and the system was solvated with SPC (simple point charge) water model. Using the genion module inherent to GROMACS, the overall charge of the system was neutralized by adding 23 NA ions. The steepest descent minimization algorithm was utilized to carry out energy minimization of the system. Isothermal-isochoric (NVT) equilibration of the system was carried out for 100 ps with short-range electrostatic cutoff value of 1.2 nm. Then the Isobaric (NPT) equilibration of the system was carried out for 100 ps as well with short-range van der Waals cutoff fixed at 1.2 nm. Finally a 50 ns molecular dynamic simulation was done using periodic boundary conditions and time integration step of 2 fs. After every 100 ps, the energy of the system was recorded. The Particle Mesh Ewald (PME) method was employed for calculating the long range electrostatic potential. The short-range van der Waals cutoff was set to 1.2 nm. The simulation temperature was maintained using modified Berendsen thermostat while the pressure was made constant using the Parrinello-Rahman algorithm. An interval of 100 ps was used each snapshot for analyzing the trajectory data. Eventually the trajectory information gathered throughout the simulation were concatenated to calculate and plot root mean square deviation (RMSD), root mean square fluctuation (RMSF), radius of gyration (Rg) and solvent accessible surface area (SASA) data. MD simulations were performed on the "bioinfo-server" running on Ubuntu 18.4.5 operating system located at the Bioinformatics Division, National Institute of Biotechnology.

In order to evaluate structural stability, Root Mean Square Deviation (RMSD) calculation was performed. The "rms" module built into the GROMACS software was utilized to extract RMSD information throughout the course of the simulation. The result can be plotted graphically using the Xmgrace package.

Root Mean Square Fluctuation (RMSF) measurement was used to determine the flexibility of local structures within the ACE2-Spike protein complex. The higher RMSF values corresponded to higher flexibility of a region. RMSF calculations were carried out using the "rmsf" module and the figures were generated using Xmgrace.

To determine the degree of compactness, the radius of gyration of the complex was calculated. A relatively steady value of radius of gyration means stable folding of a protein. Fluctuation of radius of gyration implies the unfolding of the protein. The "gyrate" module was used to generate the radius of gyration graphs for our proteins.

Hydrophobic interactions composed of non-polar amino acids are crucial for maintaining the stability of the hydrophobic core of proteins. They do so by covering the non-polar amino acids within the hydrophobic cores and keeping them at a distance from the solvent. Solvent Accessible Surface Area (SASA) is used in molecular dynamic simulations to predict the hydrophobic core stability of proteins. In this study, SASA was calculated using the "sasa" module and the resulting graph was visualized using Xmgrace.

## Results

### SARS-CoV-2 genomes from Bangladesh

From the first instance of SARS-CoV-2 genome submission from Bangladesh (May 12, 2020) to the time of the present study (June 6, 2021), the GISAID database recorded 1569 SARS-CoV-2 isolates from Bangladesh. According to our analysis, a total of 1074 samples belonged to wave-1 and 495 samples to wave-2. The length of the genomes ranges between 29,009 and

29,971 nucleotides in samples from wave-1, whereas wave-2 samples had genome lengths ranging from 29,148 to 29,854 nucleotides.

## Frequency of SARS-CoV-2 mutations in Bangladesh

In comparison to the Wuhan reference sequence, all sequences from both waves appeared to have two or more mutations (**S5 File**). The average number of mutations per sample was found to differ significantly between the two waves (**Fig 1**) based on two sided t-test $p = 2.2 \times 10^{-16}$. The average rate of mutation in wave-2 samples (30.79%) was substantially higher than the wave-1 samples (12.32%). In the case of wave-1 (1074 samples), most of the sequences possessed 6 to 17 mutations per sample, while the majority of the sequences in wave-2 (495 samples) tended to have 28 to 38 mutations (**Fig 2**).

## Type of SARS-CoV-2 mutations in Bangladesh

The occurrence of several classes of mutations, as well as the percentages of each class for both waves, are documented in the **S6 File**. Single-nucleotide polymorphisms (SNPs) seemed to be highly prevalent in both cases (58.89% in wave-1 and 61.5% in wave-2) (**Fig 3**). Extragenic mutations were also found to some extent, but all were either in 5'-UTR or in 3'-UTR regions. Wave-2 cases (943 events, 6.19%) had considerably more deletion events than wave-1 instances (115 events, 0.87%). During the first surge of the pandemic, the insertion (0.03%), deletion_stop(0.01%), or insertion_stop(0.01%) events took place in a small fraction of cases, but not at all in the second phase.

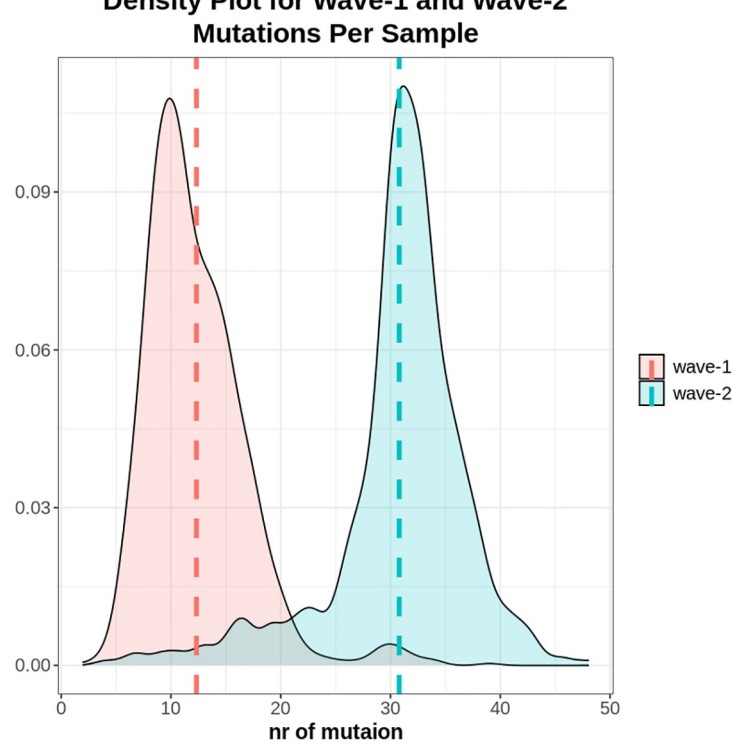

**Fig 1. Density plot of mutations per sample in case of wave-1 and wave-2.** The Red line and Blue line represent the average value of mutation per sample for wave-1 and wave-2 respectively. Wave-2 samples generally possessed a higher number of mutations per sample than wave-1.

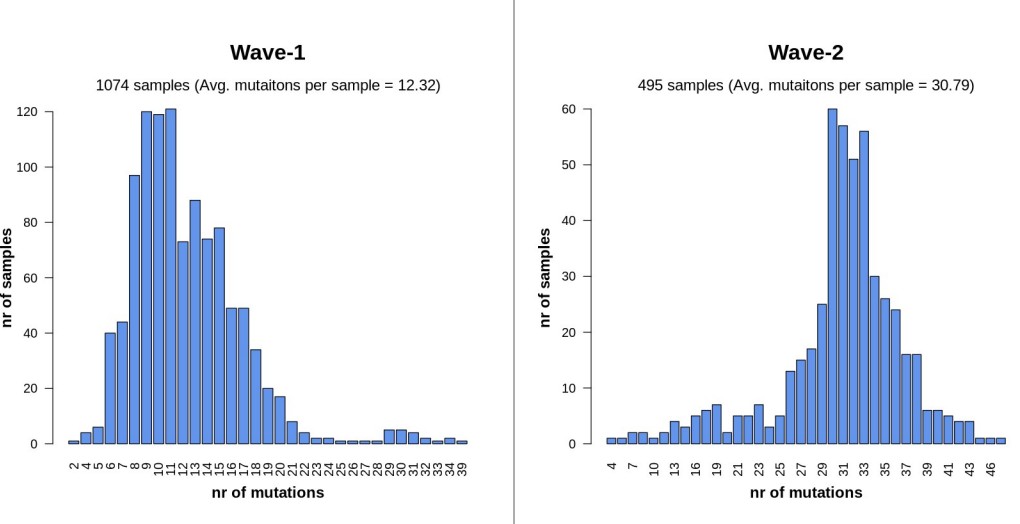

**Fig 2. Number of mutations per sample for wave-1 and wave-2.** The majority of sequences from wave-1 had 6 to 17 mutations per sample. On the other hand, most of the samples of wave-2 possess on average 28 to 38 mutations.

All mutational events were also classified into different variant types to explain the higher frequency of SNPs (**S7 File**). Though both SNP transitions (purine to purine, pyrimidine to pyrimidine) and transversion (purine to pyrimidine and vice versa) were observed among all samples, C to T transition was the most frequent mutation in both waves (**Fig 4**). The percentage of occurrence of this transitional event was 45.75% in wave-1 and 42.44% in wave-2. While A to G transition is the second most common event in wave-1(12.53%), G to T transversion possessed this place in the case of wave-2(13.57%). Oligonucleotide deletion was also commonly present in the samples from wave-2. In the second wave of the COVID-19 pandemic in Bangladesh, two oligonucleotide deletion events (`TCTGGTTTT` and `CTTGCTTTA`) appeared to be much more pervasive (2.46% and 1.63% respectively).

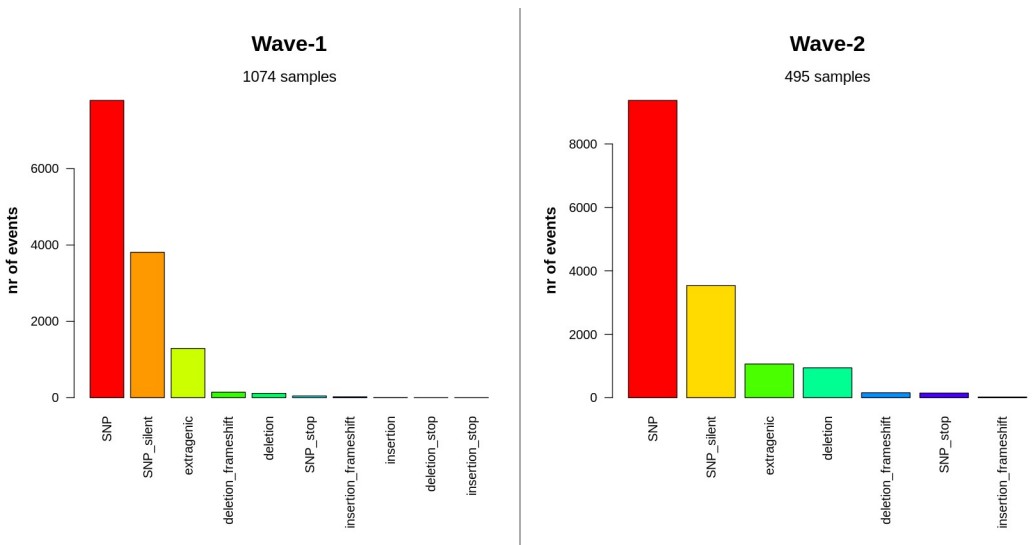

**Fig 3. Distribution of mutation classes between two waves of COVID-19 in Bangladesh.** SNPs, silent SNPs, and extragenic mutations were similarly abundant in both waves. But, sequences of wave-2 had a significant amount of deletion events.

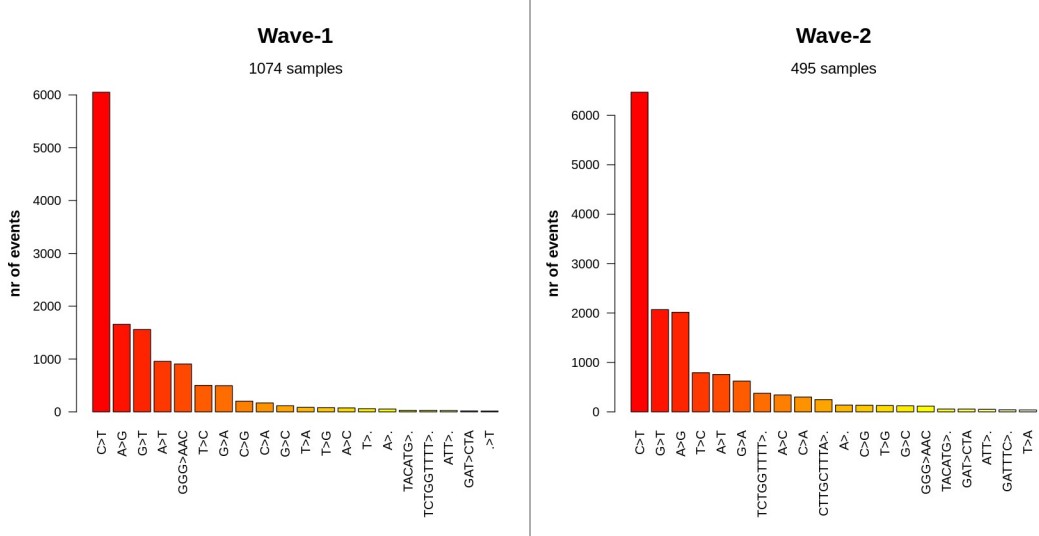

**Fig 4. Most frequent mutational events per type between the two waves.** The C to T transition was highly prevalent in both waves. While A to G transition is the second most abundant event in wave-1, samples from wave-2 had G to T transversion event.

## Genomic location of the SARS-CoV-2 mutations

The presence of mutational changes in specific coordinates of the SARS-CoV-2 genome sequences was also analyzed in this study (**S8 File**). In both waves, the A23403G, C3037T, C14408T, and C241T mutations showed a similar pattern of abundance (**Fig 5**). Although the GGG28881AAC trinucleotide substitution was the 5th most prevailing event in the case of Wave-1, its existence was much lower in the case of Wave-2 (only 0.75%). Rather TCTGGTTTT11288 deletion was substantially more common in the second phase, which is consistent with previous findings of this study.

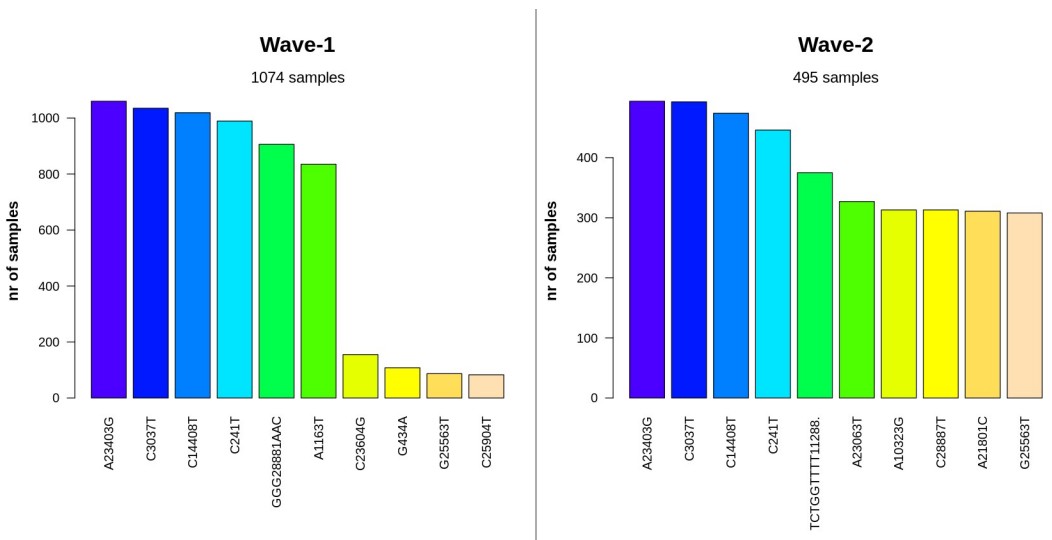

**Fig 5. Most frequent mutations at the nucleotide level in wave-1 and wave-2.** The first four nucleotide events were the most widespread for both waves in Bangladesh. Among them, A23403G, C3037T, and C241T mutations are characteristic features of G clade and its derivatives. Wave-2 showed a substantial frequency of the TCTGGTTTT oligonucleotide deletion at 11288 position.

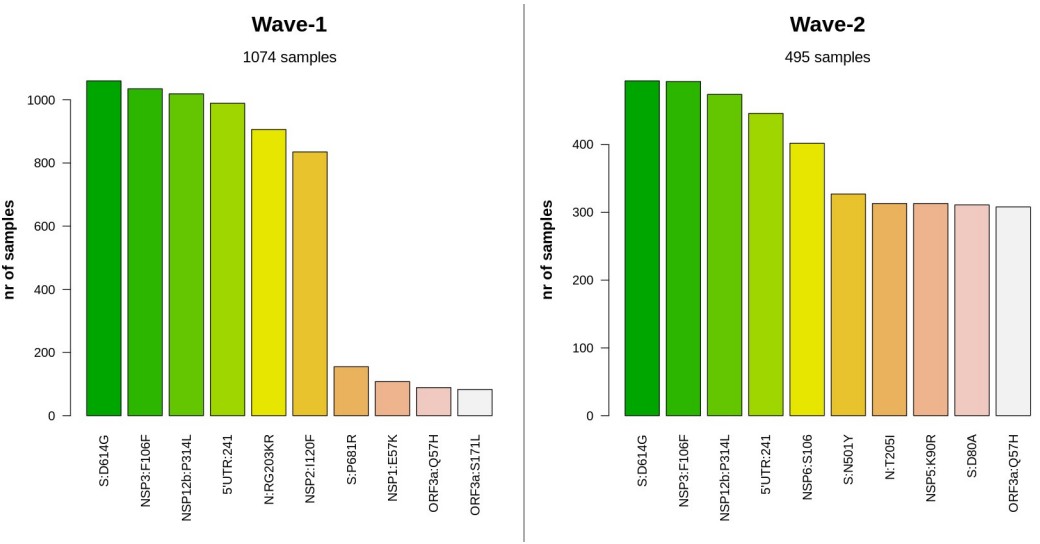

**Fig 6. Number of most frequent protein level alterations among two waves in Bangladesh.** The four most frequent amino acid substituting events had the same type of distribution for both waves. Besides, some mutations in spike and nucleocapsid protein (e.g., S:N501Y, N:T205I, S:D80A) were found in a significant amount in wave-2. Furthermore, the ORF3a:Q57H variant, which is a marker variant for GH clade, was also very common in this case.

## Impact of mutations on the SARS-CoV-2 proteome

In this step of mutational investigation we summarized the impacts of these mutations on the protein sequence of SARS-CoV-2 (**S9 File**). The D614G (aspartate to glycine in the 614th amino acid) mutation in the spike protein of SARS-CoV-2 is caused by the most predominant nucleotide transversion (A to G) in the 23,403rd position. This mutation, a characteristic feature of the G-clade of SARS-CoV-2 genome, was observed in the highest frequency in the samples of both waves (**Fig 6**). From this observation it can be said that the G-clade of this virus was ubiquitous in Bangladesh in the case of both waves, which we have also explicated later through the clade distribution analysis. Despite sorting by frequency, the pattern of dominance of S:D614G, NSP3:F106F, NSP12b:P314L and, 5'UTR:241 mutations was identical in both waves, however in wave-1, these mutations were at a somewhat greater percentage (8.01%, 7.82%, 7.70% and, 7.48% respectively) than other modifications. On the other hand, these mutations accounted for 3.24%, 3.22%, 3.11%, and 2.93% of total amino acid alteration events from wave-2 samples.

## Distribution of SARS-CoV-2 clades in Bangladesh

Furthermore, the distribution of various SARS-CoV-2 clades and most frequent variants was also compared across two waves in Bangladesh (**S10 File**). Throughout the pandemic in Bangladesh, the G-clade and its derivatives (GH, GV, GR, GRY) continued to be dominant (**Fig 7**). Although the GR clade was predominant during wave-1(75.86%), in wave-2 the GH clade took the lead (61.26%). However, the percentage of other G-clades was pretty much similar in both phases of the pandemic. On the contrary, in wave-1, the L, O, and S clades had a very low frequency, and in wave-2, the L and S clades disappeared. The variants from the B lineage were extremely common in wave-1, with B.1.1.25 accounting for 72.46% of the total (**Fig 8**). Besides, the alpha variant (B.1.1.7), a variant of concern, also showed up to some extent. In the scenario of wave-2, however, the B.1.351.3 (57.44%) variant dominated throughout the entire time frame. During this wave, there was also a progressive increase of VOC variants (alpha, beta, delta).

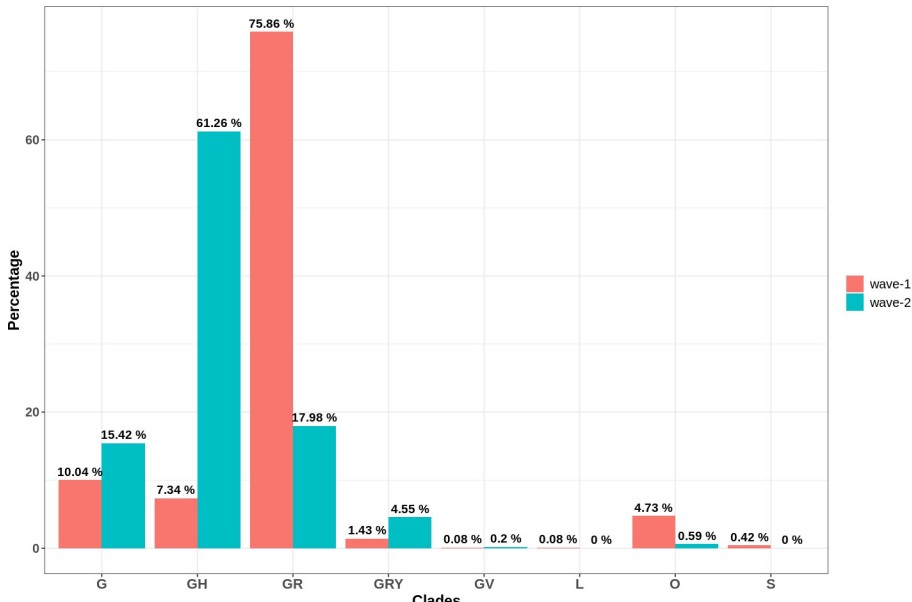

**Fig 7. Comparative distribution of different clades in two waves of COVID-19 pandemic in Bangladesh.** G clade and its descendants dominated throughout the pandemic. But GR clade in the case of wave-1 and GH clade in the case of wave-2 were the most prevalent ones. In the second wave, the L, O, and S clades were almost completely lost.

## Selection pressure and phylogenetic analysis

During transition from wave-1 to wave-2 effect of selection pressure on SARS-CoV-2 was observed mainly on NSP6, S, N, NSP5, ORF3a proteins (**S9 File**). When compared to wave-1, almost all the top mutations accumulated on the spike protein seemed to have changed dramatically during wave-2. Except S:D294D, S:Q675H, S:L5F mutations rest of the top mutations increased notably (**Fig 9**). The frequency of conversion from proline to arginine at 681

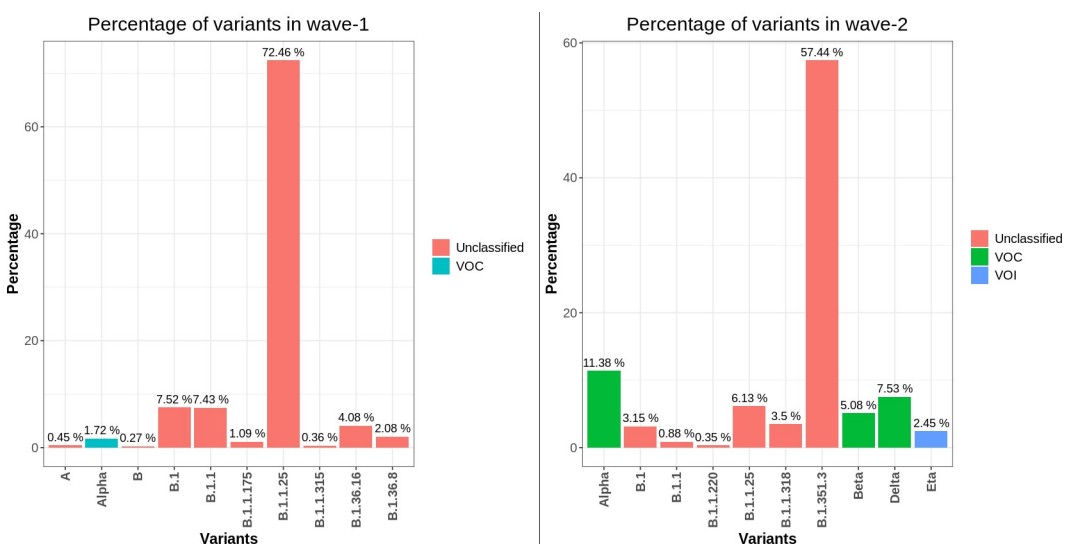

**Fig 8. Prevalence of different variants (in percentage) observed in wave-1 and 2 of COVID-19 pandemic in Bangladesh.** Almost all SARS-CoV-2 variants of wave-1 were from B lineage. On the other hand, wave-2 had an increased number of VOC variants. The most prevalent B.1.351.3 variant of this wave is a derivative of the beta (B.1.351) variant.

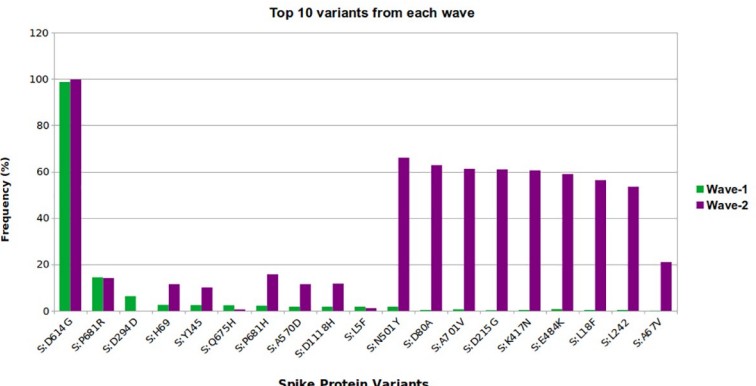

**Fig 9. Selection pressure of SARS-CoV-2 spike protein variants in wave-1 and wave-2.** Between two waves of the COVID-19 pandemic in Bangladesh, SARS-CoV-2 demonstrated a different pattern of spike protein variation as a result of selection pressure. Most of the top mutations appear to be becoming more pervasive with time.

position of spike protein were almost the same in both wave. However another type of conversion (proline to histidine) at the same position followed similar pattern of increment as with the other mutations. Three mutations namely S:K417N, S:E848K and, S:N501Y were of particular interest since they belong to the receptor binding domain of spike protein of SARS-CoV-2. The molecular phylogeny demonstrates that the SARS-CoV-2 genomes from Bangladesh clustered in different distinct clades of VOCs and VOIs (**S1 Fig**).

## Molecular docking between the ACE2 and spike protein

Molecular docking between human ACE2 receptor and the receptor binding domain (RBD) of Spike protein found in the reference and the 3 most commonly found variants namely, B.1.1.7, B.1.351, B.1.617 revealed that the variants B.1.351 and B.1.617 had the two highest binding affinities respectively. Notably, the variant B.1.617 exhibited the highest docking score (**Table 1**).

## Molecular dynamic simulation of the ACE2-spike protein complex

Protein backbone RMSD analysis of reference spike protein and the variants exhibited marked differences. The reference protein periodically showed large deviations until it attained stability at around 42 ns. The variant B.1.1.7 was much more stable. Despite some initial fluctuations it assumed stable conformation gradually after 20 ns. B.1.351 on the other hand was very stable since the beginning. However there were a number of spikes in between 34 and 40 ns after which the complex stabilized again. Among the four complexes tested, the one involving the variant B.1.617 was the most stable of all. It remained highly stable throughout the simulation except there was a rise in RMSD at 36–46 ns period. However it maintained a steady value within this period as well (**Fig 10A**).

**Table 1. Results of molecular docking between SARS-CoV-2 spike glycoprotein Receptor Binding Domain (RBD) and human ACE2 receptor.** The binding affinity was measured in physiological temperature (37˚C).

| Spike protein variant ACE2 | TongDock_A docking score | TongDock_A Cluster size | Binding affinity, ΔG (kcal mol-1) | Kd (M) at 37.0˚C |
|---|---|---|---|---|
| Reference | 1262.964 | 34 | -13.4 | 3.5E-10 |
| B 1.1.7 | 1293.348 | 36 | -12.8 | 1.0E-09 |
| B.1.351 | 1447.344 | 32 | -16.4 | 2.9E-12 |
| B.1.617 | 1541.675 | 39 | -16.2 | 3.5E-12 |

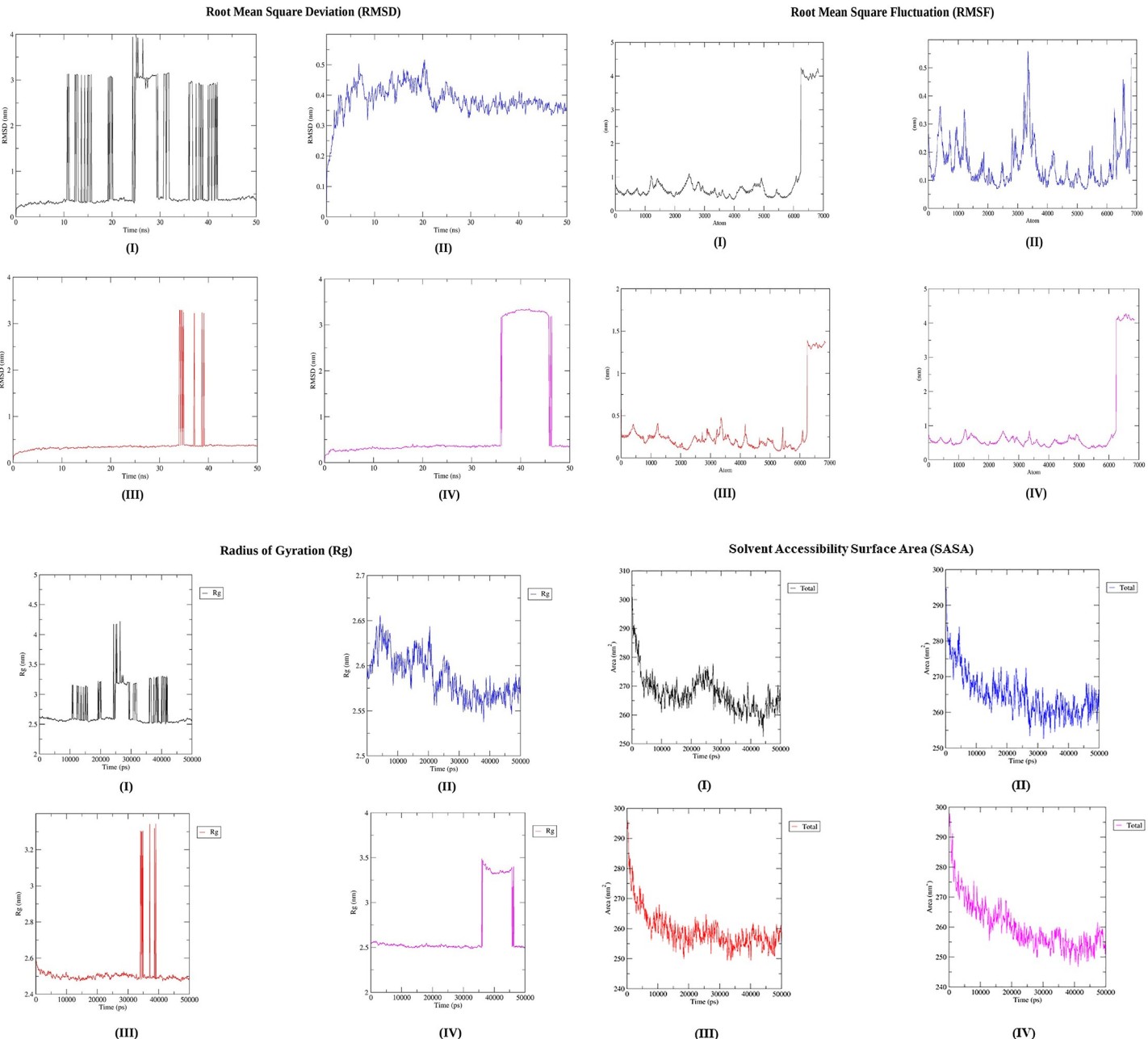

**Fig 10. A.** RMSD analysis of (I) Reference-ACE2 (II) B.1.1.7-ACE2 and (III) B.1.351-ACE2 complex (IIV) B.1.617-ACE2 complex. The last one appeared to be most stable among all. **B.** RMSF values of (I) Reference-ACE2 (II) B.1.1.7-ACE2 and (III) B.1.351-ACE2 complex (IIV) B.1.617-ACE2 complex. Except B.1.1.7, the rest of the variants showed very regional flexibility. **C.** Rg measurement of (I) Reference-ACE2 (II) B.1.1.7-ACE2 and (III) B.1.351-ACE2 complex (IIV) B.1.617-ACE2 complex. The B.1.617-ACE2 complex remained more compact than the rest. **D.** Solvent Accessibility Area calculation of (I) Reference-ACE2 (II) B.1.1.7-ACE2 and (III) B.1.351-ACE2 complex (IIV) B.1.617-ACE2 complex. Although solvent accessibility area gradually declined gradually for all 4 complex, the B.1.617-ACE2 complex lost solvent accessibility area the most.

Upon evaluation of the detailed residual atomic fluctuations through RMSF calculation of the protein Cα atoms, it was apparent that both the reference and the variants B.1.351, and B.1.617 were very similar in the sense that the atoms near the end of the complex were more flexible than the rest of the complex (**Fig 10B**).

The Rg graphs revealed a very similar pattern found in the RMSD graphs. The variant B.1.617 remained in compact state throughout the simulation with a period of unfolding at 36–46 ns. The reference protein unfolded at regular intervals and often to a high degree. The variant B.1.1.7 folded steadily while B.1.351 unfolded abruptly several times from 34–40 ns and remained otherwise rest of the time (**Fig 10C**).

SASA values provided a measure of the complex's susceptibility to disruption of their hydrophobic core by water. For all four complexes, the SASA declined gradually over time. However, the greatest reduction took place in the case of the variant B.1.617 (**Fig 10D**).

## Discussion

The first case of COVID-19 in Bangladesh was identified on migrants returning from Italy at the beginning of March [70]. Since then, according to the official record (as of Jun 15, 2021) about 829K people have been affected and over 13K died of this virus in Bangladesh. Meanwhile, the virus has affected over 177 million individuals globally, with over 3.8 million deaths [1]. Since the first patient had been identified, COVID-19 cases were found regularly in Bangladesh throughout the year 2020. The number of COVID-19 cases in neighboring India has been rising rapidly since March 2021. Inevitably, the number of SARS-CoV-2 cases in Bangladesh is also on the upswing. In this study, a comparative genomic analysis was performed to track the dynamics of SARS-CoV-2 evolution between the two waves of the COVID-19 pandemic in Bangladesh.

The rate of mutation in SARS-CoV-2 (~2 nucleotides/month) is far lower than that of influenza (4 nucleotides/month) or HIV (8 nucleotides/month), yet its distinct genomic regions and proteins are mutating at significantly variable rates [71, 72]. The frequency of these mutations alters considerably depending on the geographical location with time as well. In this study, a significant rise in the rate of mutation was observed in wave-2 samples of Bangladesh compared to the wave-1. A similar pattern was observed in the instance of SARS-CoV-2 pandemic waves in Hiroshima, Japan [73]. Although both waves in Bangladesh had a higher incidence of amino acid altering SNPs, wave-2 tended to have a higher number of deletion events (**Fig 3**). Such recurring recurrent deletion events in the SARS-CoV-2 genome had been reported to facilitate its transmission with altered antigenicity and antibody escape mechanism [74]. Furthermore, despite the fact that C to T transitions prevailed in both waves, G to T transversion was rather frequent in wave-2 (**Fig 4**). This transversion provoked the G25563T nucleotide mutation event (ORF3a:Q57H in protein level) in the SARS-CoV-2 genome, which was a marker variant for GH clade [75]. This might explain why the GH clade was observed to be more apparent in Bangladesh during Wave 2. Different marker variants for GISAID clades and lineages have been listed in **Table 2**. The phylogenetic clusters derived from the statistical distribution of SARS-CoV-2 genomic distances have been used to define these clade classifications in GISAID [76].

The second and third most common SNPs in Bangladesh were silent and extragenic SNPs, respectively. Even though these SNPs do not alter the protein sequence directly, they have a major impact on the efficiency of translation and transcription. SNPs in the 5'-UTR, in particular, can influence the virus's transcription and replication processes by altering the folding of genomic RNA [76]. The A23403G, G3037T, C241T nucleotide variants as well as S: D614G, ORF3a:Q57H, N:G204R protein variants were equally abundant in both waves (**Fig 6**). All of these are marker variants for the G clade and its derivatives (**Table 2**), which explains why Bangladesh experienced a greater distribution of these clades (**Fig 7**).

In May 2021, the World Health Organization (WHO) recommended adopting Greek Alphabet letters to name several important SARS-CoV-2 variants (**Table 3**). The most widely

Table 2. List of the marker variants for GISAID clade and lineage [77].

| GISAID Clade | Lineage | Nucleotide Events | Protein Events |
|---|---|---|---|
| S | A | C8782T, T28144C | NS8:L84S |
| L | B | Reference Genome from Wuhan | |
| V | B.2 | G11083T, G26144T | NSP6:L37F, NS3:G251V |
| G | B.1 | C241T, C3037T, A23403G | S:D614G |
| GH | B.1.* | C241T, C3037T, A23403G, G25563T | S:D614G, ORF3a3:Q57H [A17] |
| GR | B.1.1 | C241T, C3037T, A23403G, G28882A | S:D614G, N:G204R |
| GV | B.1.177 | C241T, C3037T, A23403G, C22227T | S:D614G, S:A222V |
| GRY | B.1.7 | C241T, C3037T, 21765-21770del, 21991-21993del, A23063T, A23403G, G28882A | S:H69del, S:V70del, S:Y144del, S:N501Y, S:D614G, N:G204R |

available B.1.351.3 variant in wave-2 in Bangladesh is a sublineage of beta, a VOC variant first detected in South African samples. On the other hand, the delta variant, which is driving a catastrophic pandemic in neighboring country India, is the third most frequent variant in this wave in Bangladesh (Fig 8).

From our selection pressure and molecular phylogeny analysis, rise of new mutations was observed during transition from wave-1 to wave-2 (Fig 9). Here, B 1.1.25 variant was the most predominant in the first wave but in the ending phase of the first wave presence of the Alpha variant was observed in the Nextstrain phylogenetic tree (S1 Fig). After that, the Eta and Delta variants entered around February 2021 and April 2021 respectively. These scenarios can be explained by the mutations in spike protein. In case of first wave, several SARS-CoV-2 variants tried to increase their transmissibility through certain spike protein mutations. For example, mutations such as S:P681R/H were associated with enhanced transmission [79]. It has been reported that variants with S:D614G and S:Q675H mutations are relatively less infectious [80]. However, in the second wave, various SARS-CoV-2 VOIs and VOCs from different countries found their way into Bangladesh. These VOIs and VOCs had several mutations in spike receptor binding domain (e.g., SN501Y, SK417N, SE484K) which made the virus more infectious [81]. Such mutations have been found to escape the immune system (both vaccinated/ non-vaccinated) and spread faster [82, 83]. Therefore, we are suggesting that these mutations in the spike protein RBD made the variants of wave-2 more predominant as a consequence of Darwinian natural selection.

The importance of the interaction between the ACE2 receptor and the SARS-CoV-2 spike protein is paramount in understanding the pathogenesis of COVID-19 infection [84].

Table 3. Naming SARS-CoV-2 variants by World Health Organization (WHO) [78].

| WHO Label | Lineage | Variant Class | First Detected Samples |
|---|---|---|---|
| Alpha | B.1.1.7 | VOC | UK, Sep-2020 |
| Beta | B.1.351 | VOC | South Africa, May-2020 |
| Gamma | P.1 | VOC | Brazil, Nov-2020 |
| Delta | B.1.617.2 | VOC | India, Oct-2020 |
| Epsilon | B.1.427, B.1.429 | VOI | USA, Mar-2020 |
| Zeta | P.2 | VOI | Brazil, Apr-2020 |
| Eta | B.1.525 | VOI | Several Countries, Dec-2020 |
| Theta | P.3 | VOI | Philippines, Jan-2021 |
| Iota | B.1.526 | VOI | USA, Nov-2020 |
| Kappa | B.1.617.1 | VOI | India, Oct-2020 |

Molecular docking and dynamics simulations are one of the key tools for getting in-depth insights into these interactions [85–87].

Molecular docking experiment showed that the Spike protein of the variants B.1.351 and B.1.617 bound most tightly to the ACE2 receptor. From the 50 ns molecular dynamics simulation carried out in GROMACS, the complex between the ACE2 receptor and the Spike protein variant of B.1.617 was found to be most stable considering its structural deviation, local area flexibility, state of folding, and susceptibility to disruption by solvents (**Fig 10**). These findings coincide with those reported in other studies and the enhanced Spike protein stability of the B.1.617 is likely to contribute to the efficiency of transmission of SARS-CoV-2 [88–90].

## Conclusions

The current study revealed significant genomic and proteomic differences in the SARS-CoV-2 viral isolates circulating within the perimeters of Bangladesh between the first and the second wave of the COVID-19 pandemic. They differ in terms of distribution of clades, mutations, variants, rate of mutations, and even in terms of their interactions with the host ACE2 receptor. The study found evidence that the B.1.617 lineage of the virus is likely to be more infectious than others. Notably, any existence of a domestic variant is yet to be detected. Therefore, if Bangladesh can shield itself from the arrival of SARS-CoV-2 variants from outside for a substantial period, the COVID-19 pandemic in the country is likely to come to an end.

## Supporting information

**S1 File. Sample info of wave-1.** GISAID information about samples from wave-1.
(PDF)

**S2 File. Sample info of wave-2.** GISAID information about samples from wave-2.
(PDF)

**S3 File. Ref annotation.** GFF3 annotation file of the reference sequence.
(GFF)

**S4 File. Full report.** Detailed report on the mutation analysis of wave-1 and wave-2.
(XLSX)

**S5 File. Mutation number in samples.** Frequency of mutations in wave-1 vs wave-2.
(XLSX)

**S6 File. Mutation classes.** Types of mutations found in wave-1 and wave-2.
(XLSX)

**S7 File. Variant types.** Types of variants found in wave-1 and wave-2.
(XLSX)

**S8 File. Nucleotide events.** Types of mutations at the nucleotide level.
(XLSX)

**S9 File. Protein events.** Types of mutations at the nucleotide level.
(XLSX)

**S10 File. Variant and clades distribution.** Distribution of SARS-CoV-2 variants and clades in Bangladesh.
(XLSX)

**S1 Fig. Molecular phylogeny.** Molecular phylogeny of SARS-CoV-2 genomes from Bangladesh.
(PNG)

## Author Contributions

**Conceptualization:** Ishtiaque Ahammad, Mohammad Uzzal Hossain, Md. Salimullah.

**Formal analysis:** Ishtiaque Ahammad, Anisur Rahman, Zeshan Mahmud Chowdhury, Arittra Bhattacharjee.

**Investigation:** Ishtiaque Ahammad, Anisur Rahman, Zeshan Mahmud Chowdhury, Arittra Bhattacharjee.

**Methodology:** Anisur Rahman, Arittra Bhattacharjee.

**Project administration:** Ishtiaque Ahammad, Mohammad Uzzal Hossain.

**Software:** Anisur Rahman.

**Supervision:** Mohammad Uzzal Hossain, Keshob Chandra Das, Chaman Ara Keya, Md. Salimullah.

**Validation:** Ishtiaque Ahammad.

**Visualization:** Ishtiaque Ahammad, Anisur Rahman.

**Writing – original draft:** Ishtiaque Ahammad, Anisur Rahman, Zeshan Mahmud Chowdhury, Arittra Bhattacharjee.

**Writing – review & editing:** Ishtiaque Ahammad, Anisur Rahman, Zeshan Mahmud Chowdhury, Md. Salimullah.

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
