## [Decision Letter · Decision Letter 0]

29 Jul 2021

PONE-D-21-19985

Comparative Genomic Study for Revealing the Complete Scenario of COVID-19 Pandemic in Bangladesh

PLOS ONE

Dear Dr. Ahammad,

Thank you for submitting your manuscript to PLOS ONE. After careful consideration, we feel that it has merit but does not fully meet PLOS ONE’s publication criteria as it currently stands. Therefore, we invite you to submit a revised version of the manuscript that addresses the points raised during the review process.

We look forward to receiving your revised manuscript.

Kind regards,

Narasimha Murthy Bhamidipati, Ph.D

Academic Editor

PLOS ONE

2. Please note that in order to use the direct billing option the corresponding author must be affiliated with the chosen institute. Please either amend your manuscript to change the affiliation or corresponding author, or email us at plosone@plos.org with a request to remove this option.

Reviewers' comments:

Reviewer's Responses to Questions

**Comments to the Author**

1. Is the manuscript technically sound, and do the data support the conclusions?

Reviewer #1: Yes

Reviewer #2: Yes

2. Has the statistical analysis been performed appropriately and rigorously? 

Reviewer #1: No

Reviewer #2: Yes

3. Have the authors made all data underlying the findings in their manuscript fully available?

Reviewer #1: Yes

Reviewer #2: Yes

4. Is the manuscript presented in an intelligible fashion and written in standard English?

Reviewer #1: Yes

Reviewer #2: Yes

5. Review Comments to the Author

Reviewer #1: The manuscript “Comparative Genomic Study for Revealing the Complete Scenario of COVID-19 Pandemic in Bangladesh” (PONE-D-21-19985) reports the SARS-CoV2 variants that have caused first and second wave in Bangladesh. The authors have compared the causative variants responsible for the two waves. The manuscript seems to be the first report detailing the molecular perspectives of the two consecutive waves of COVID19 infection in Bangladesh. The evolutionary perspectives of the viral genome has been studied. However, there are some queries which must be addressed by the authors to improve the manuscript.

Abstract: OK.

Introduction:

Page 8; Line: 31-32: Please clearly state based on analyses of which molecular sequences you are stating that “it was discovered that the wave-2 samples had a significantly greater average rate of mutation/sample (30.79%) than the wave-1 samples (12.32%)”

Materials and Methods:

Page 16, Line 208: Please correct “Room Mean Square Fluctuation (RMSF)” to “Root Mean Square Fluctuation (RMSF)”.

Results and Discussion:

1. Please specify the length of genomes that have been studied

2. What was the quality of sequencing (base calling)?

3. Compare the genomic constitution of the two waves or other specific variants.

4. The Spike protein is variable and have been reported to experience selection pressure (doi: 10.1016/j.heliyon.2020.e05001; DOI: 10.3390/pathogens9100829). Please report in the light of the samples of Bangladesh.

5. Please include elaborated analysis on selection pressure and the molecular phylogeny of the variants for the genes showing variation among the viral variants.

English: The formation of sentences is not correct. There are several grammatical and syntactical errors. The English of the manuscript must be get thoroughly checked by a suitable English language professional. Some of the points have been indicated in the attached manuscript.

Please don’t use the symbols like “>”, in running text, rather write “more than” or “greater than”

Reviewer #2: The manuscript presented falls within the scope of the Journal and adds knowledge about the distribution of various clades of SARS-CoV-2 in Bangladesh. The manuscript is well written with enough data both in terms of extensiveness and representation.

However there are certain minor corrections in the manuscript which can be found in pop-up notes of the attachment. I also suggest you to modify the title as indicated in the pop-up note of the title.

6. PLOS authors have the option to publish the peer review history of their article (what does this mean?). If published, this will include your full peer review and any attached files.

Reviewer #1: **Yes: **CS Mukhopadhyay

Reviewer #2: **Yes: **KAMISETTY ASWANI KUMAR

---

## [Author Response · Author response to Decision Letter 0]

1 Sep 2021

# Response to Queries from Reviewer 1

Reviewer’s Query:

The manuscript “Comparative Genomic Study for Revealing the Complete Scenario of COVID-19 Pandemic in Bangladesh” (PONE-D-21-19985) reports the SARS-CoV2 variants that have caused first and second wave in Bangladesh. The authors have compared the causative variants responsible for the two waves. The manuscript seems to be the first report detailing the molecular perspectives of the two consecutive waves of COVID19 infection in Bangladesh. The evolutionary perspectives of the viral genome has been studied. However, there are some queries which must be addressed by the authors to improve the manuscript.

Abstract: OK.

Introduction:

Page 8; Line: 31-32: Please clearly state based on analyses of which molecular sequences you are stating that “it was discovered that the wave-2 samples had a significantly greater average rate of mutation/sample (30.79%) than the wave-1 samples (12.32%)”

Authors’ Response

Abstract/Introduction

1. Information about the type of molecular sequences has been added into the abstract which can be found at page 2, line 30-33.

Reviewer’s Query:

Materials and Methods:

Page 16, Line 208: Please correct “Room Mean Square Fluctuation (RMSF)” to “Root Mean Square Fluctuation (RMSF)”.

Authors’ Response

Materials and Methods

“Room Mean Square Fluctuation (RMSF)” has been changed to “Root Mean Square Fluctuation (RMSF)” at page 10, line 222.

Reviewer’s Query:

Results and Discussion:

1. Please specify the length of genomes that have been studied

2. What was the quality of sequencing (base calling)?

3. Compare the genomic constitution of the two waves or other specific variants.

4. The Spike protein is variable and have been reported to experience selection pressure (doi: 10.1016/j.heliyon.2020.e05001; DOI: 10.3390/pathogens9100829). Please report in the light of the samples of Bangladesh.

5. Please include elaborated analysis on selection pressure and the molecular phylogeny of the variants for the genes showing variation among the viral variants.

English: The formation of sentences is not correct. There are several grammatical and syntactical errors. The English of the manuscript must be get thoroughly checked by a suitable English language professional. Some of the points have been indicated in the attached manuscript.

Please don’t use the symbols like “>”, in running text, rather write “more than” or “greater than”

Authors’ Response

Result and Discussion

1. Length of the genomes has been added at page 11, line 241-243.

2. The study was performed on assembled genomes from GISAID database. GISAID does not contain raw reads or quality score of the sequences. However, we only chose complete genomes for our study.

3. Comparison of genomic constitution between wave-1 and wave-2 has been provided in the following paragraphs;

3.2 Frequency of SARS-CoV-2 Mutations in Bangladesh

3.3 Type of SARS-CoV-2 Mutations in Bangladesh

3.4 Genomic Location of the SARS-CoV-2 Mutations

3.6 Distribution of SARS-CoV-2 Clades and Variants in Bangladesh

4 and 5. Analysis of selection pressure and changes in the variability in spike protein have been calculated and discussed along with molecular phylogeny of the variants. The newly added parts in this regard can be found in the following sections of the manuscript-

Material methods section

2.4 Analysis on Selection Pressure and Molecular Phylogeny of The Variants in page 8-9, line 173-180.

Result section

3.7 Selection Pressure and Phylogenetic Analysis in page 16-17, line 339-355.

Discussion

The findings regarding selection pressure and phylogeny analysis are discussed in page 23-24, line 468 to 483.

Figure and Supplementary file

Newly added Figures and Supplementary files-

Fig 9 in page 47, line 815

Supplementary file 11

English:

 • “>” sign previously used to denote mutations has been replaced with “to” in running text.

 • Grammatical adjustments as highlighted in the honorable reviewer's attachment has been carried out. The whole of the manuscript was also checked for grammatical errors and corrected wherever needed.

# Response to Queries from Reviewer 2

Reviewer’s Query:

The manuscript presented falls within the scope of the Journal and adds knowledge about the distribution of various clades of SARS-CoV-2 in Bangladesh. The manuscript is well written with enough data both in terms of extensiveness and representation.

However there are certain minor corrections in the manuscript which can be found in pop-up notes of the attachment. I also suggest you to modify the title as indicated in the pop-up note of the title.

Authors’ Response

We thank the honorable reviewer #2 for his suggestions for the title, considered it seriously and felt the need to modify the title slightly to better represent the contents of the study. The new title is as following- “Wave-wise comparative genomic study for revealing the complete scenario and dynamic nature of COVID-19 pandemic in Bangladesh”. We did not include the word “clade-wise” in the title because the study was not just based on distribution of various clades of SARS-CoV-2 in Bangladesh but a complete scenario of the pandemic from a comparative genomics perspective. For example, we analyzed the distribution of mutations per sample, classes of mutations, most frequent mutational events, distribution of different clades, prevalence of different variants, selection pressure and phylogenetic analysis between wave-1 and wave-2 as well as the effect of mutations on the transmissibility of the virus. Honorable reviewer #2 also suggested to include the phrase “Genomic Surveillance” in the title. Even though our study encompassed genomic surveillance, it was more than that. Our main focus was on the comparison of genomic features of SARS-CoV-2 between the first and the second wave in Bangladesh. Therefore, we used “Wave-wise comparative genomic study” instead of “Genomic Surveillance” in the title to denote the nature of our work.

Other minor corrections suggested by honorable reviewer #2 has also been executed in the revised manuscript.

---

## [Decision Letter · Decision Letter 1]

16 Sep 2021

Wave-wise comparative genomic study for revealing the complete scenario and dynamic nature of COVID-19 pandemic in Bangladesh

PONE-D-21-19985R1

Dear Dr. Ahammad,

We’re pleased to inform you that your manuscript has been judged scientifically suitable for publication and will be formally accepted for publication once it meets all outstanding technical requirements.

Kind regards,

Narasimha Murthy Bhamidipati, Ph.D

Academic Editor

PLOS ONE

Additional Editor Comments (optional):

Reviewers' comments:

Reviewer's Responses to Questions

**Comments to the Author**

1. If the authors have adequately addressed your comments raised in a previous round of review and you feel that this manuscript is now acceptable for publication, you may indicate that here to bypass the “Comments to the Author” section, enter your conflict of interest statement in the “Confidential to Editor” section, and submit your "Accept" recommendation.

Reviewer #1: All comments have been addressed

Reviewer #2: All comments have been addressed

2. Is the manuscript technically sound, and do the data support the conclusions?

Reviewer #1: Yes

Reviewer #2: Yes

3. Has the statistical analysis been performed appropriately and rigorously? 

Reviewer #1: Yes

Reviewer #2: Yes

4. Have the authors made all data underlying the findings in their manuscript fully available?

Reviewer #1: Yes

Reviewer #2: Yes

5. Is the manuscript presented in an intelligible fashion and written in standard English?

Reviewer #1: Yes

Reviewer #2: Yes

6. Review Comments to the Author

Reviewer #1: The authors have made the required corrections/modifications. The manuscript may be accepted for publication.

Reviewer #2: (No Response)

7. PLOS authors have the option to publish the peer review history of their article (what does this mean?). If published, this will include your full peer review and any attached files.

Reviewer #1: **Yes: **CHANDRA SEKHAR MUKHOPADHYAY

Reviewer #2: **Yes: **KAMISETTY ASWANI KUMAR

---

## [Editor Report · Acceptance letter]

20 Sep 2021

PONE-D-21-19985R1 

Wave-wise comparative genomic study for revealing the complete scenario and dynamic nature of COVID-19 pandemic in Bangladesh 

Dear Dr. Ahammad:

I'm pleased to inform you that your manuscript has been deemed suitable for publication in PLOS ONE. Congratulations! Your manuscript is now with our production department. 

Kind regards, 

on behalf of

Dr. Narasimha Murthy Bhamidipati 

Academic Editor

PLOS ONE